

# The difference in expression of long noncoding RNAs in rat semen induced by high-fat diet was associated with metabolic pathways

Tian An[1,*], Hui Fan[1,*], Yu F. Liu[2], Yan Y. Pan[1], Ying K. Liu[3], Fang F. Mo[1], Yu J. Gu[1], Ya L. Sun[4], Dan D. Zhao[1], Na Yu[1], Yue Ma[1], Chen Y. Liu[1], Qiu L. Wang[3], Zheng Y. Li[1], Fei Teng[1,5], Si Hua Gao[1] and Guang J. Jiang[1]

[1] Diabetes Research Center, Beijing University of Chinese Medicine, Beijing, China
[2] Beijing University of Chinese Medicine Third Affiliated Hosiptal, Beijing, China
[3] Beijing He Ping li Hospital, Beijing, China
[4] Beijing Changping Chinese Medicine Hospital, Beijing, China
[5] State Key Laboratory of Stem Cell and Reproductive Biology, Institute of Zoology, Chinese Academy of Sciences, Beijing, China
[*] These authors contributed equally to this work.

Corresponding authors
Si Hua Gao, sihuagaopaper@sina.com
Guang J. Jiang,
guangjianjiang@sina.com

## ABSTRACT

**Background**. Obesity, a common metabolic disease, is a known cause of male infertility due to its associated health risk. Long noncoding RNAs (lncRNAs) have also been reported to be associated with male reproductive diseases; however, their role in the association between high-fat diet-induced obesity (DIO) and male reproduction remains unclear.

**Methods**. We used microarray analysis to compare the expression levels of lncRNAs and mRNAs in the spermatozoa of rats with DIO and normal rats. We selected a few lncRNAs that were obviously up-regulated or down-regulated, and then used RT-PCR to verify the accuracy of their expression. We then performed a functional enrichment analysis of the differentially expressed mRNAs using gene ontology and pathway analysis. Finally, target gene predictive analysis was used to explore the relationship between lncRNAs and mRNAs.

**Results**. The results revealed a statistically significant difference in the fasting blood glucose level in rats with DIO and control rats. We found that 973 lncRNAs and 2,994 mRNAs were differentially expressed in the sperm samples of the DIO rats, compared to the controls. GO enrichment analysis revealed 263 biological process terms, 39 cellular component terms, and 40 molecular function terms ($p < 0.01$) in the differentially expressed mRNAs. The pathway analysis showed that metabolic pathways were most enriched in protein-coding genes.

**Discussion**. To the best of our knowledge, this is the first report to show differences in the expression levels of lncRNAs and mRNAs in the sperms of rats with DIO and normal rats, and to determine the expression profile of lncRNAs in the sperm of rats with DIO. Our results have revealed a number of lncRNAs and pathways associated with obesity-induced infertility, including metabolic pathways. These pathways could be new candidates that help cope with and investigate the mechanisms behind the progression of obesity-induced male infertility.

## INTRODUCTION

Obesity is a serious, chronic metabolic disease with several comorbidities, including non-insulin-dependent diabetes mellitus, high cholesterol, heart disease, hypertension, cancer, and psychological depression (*Adams & Murphy, 2000*; *Kushner & Bessesen, 2007*). The incidence of obesity has increased rapidly in recent years, and this increase in obesity has been accompanied by a decrease in male fertility and fecundity (*Swan, Elkin & Fenster, 2000*). Obesity is known to cause serious harm to the male reproductive system, in both humans and animals (*Landry, Cloutier & Martin, 2013*). Studies have shown that obesity reduces spermatogenesis and fertility by affecting the volume, concentration, and motility of sperms. It also causes erectile dysfunction and variations in testicular tissue and sperm proteomics (*Yan et al., 2015*; *Ferramosca et al., 2016*; *Cui et al., 2017*; *Kriegel et al., 2009*; *Cabler et al., 2010*).

Long noncoding RNAs (lncRNAs) are a group of RNAs with mRNA-like transcripts, length ranging from 200 nt to 100 kb, and limited protein-coding potential (*Gibb, Brown & Wan, 2011*; *Ma, Bajic & Zhang, 2013*). Recent studies (*Matsumoto et al., 2017*) have shown that lncRNAs can be translated to encode functional peptide segments. They can participate in various biological processes, including cell cycle regulation, differentiation, and epigenetic regulation (*Wu et al., 2016*; *Rinn & Chang, 2012*). Many studies, including ours, have established an association between lncRNAs and male reproduction by revealing differences in the expression profiles of lncRNAs in the spermatozoon of mice with high-fat diet-induced obesity (DIO), diabetic mice, and normal mice (*Bao et al., 2013*). In a previous study, we had investigated the expression profile of lncRNAs in the sperm of diabetic mice; we found that 7,721 lncRNAs and 6,097 mRNAs were differentially expressed in the sperms of diabetic and normal mice (*Jiang et al., 2016*). Therefore, we hypothesized that the effects of obesity, a risk factor for diabetes, on spermatozoa were associated with lncRNA and its target genes.

In the current study, we aimed to explore the link between obesity and male infertility, based on the lncRNA expression profile. We established a rat model of obesity induced by a high-fat diet, in order to understand how it might affect the male reproductive system at the epigenetic (lncRNA) level and to explore the possible biological processes and pathways associated with obesity and associated fertility disorders.

## MATERIALS AND METHODS

### Animal models and sperm collection

All protocols in this study were approved by the Animal Care Committee of the Institute of Zoology, Chinese Academy of Sciences, and all animals were treated according to the guidelines of the Animal Care Committee. Male SD rats (6-week old; Hua Fu Kang Company, Beijing, China) were used in this study. The field experiments were approved by the Research Council of Chinese Academy of Sciences (certification number SCXK [Jing]

2011-0024). After one week of acclimation, the rats were randomly divided into DIO and normal groups. The rats in the DIO group were fed on a high-fat diet (20% sucrose, 10% lard, 2.5% cholesterol, 0.2% sodium cholic acid, and 67.3% (w/w) standard chow) for 12 weeks to induce obesity (average body weight: DIO $> (1 + 20\%)$ normal, $n = 7$). The rats in the normal group ($n = 7$) were given the standard diet. Sperm was collected from the epididymis of all the rats. Sperm collection was performed according to the method described by *Jiang et al. (2016)*. Briefly, the rats were sacrificed by cervical dislocation; the spermatozoa were extracted and placed in preheated human tubal cultures, which were then centrifuged ($1,500\times$ g, 4 °C, 8 min) to collect the supernatant fluid.

## Sperm analysis and testis histomorphology evaluation

Semen was obtained from the tail of the epididymis and quickly placed in a Centrifuge tube with 1ml of Ham's F10 medium (Bioway Biotechnology Co., Ltd., Beijing, China) for analysis of sperm motility and density using computer–assisted semen analyzer (CASA— TOX IVOS; Hamilton Thorne, Beverly, MA, USA). Testis tissue were collected from 19-week-old DIO and normal rats, and first fixed in 4% neutral formaldehyde fixative, then embded in paraffin. Sections of 4 $\mu$m thicknesses were used for hematoxylin and eosin (HE) staining, which was conducted according to convention methods. Finally, the morphological changes in testis tissue section was observed using an optical microscope (Olympus, Tokyo, Japan).

## RNA extraction

The supernatant fluid samples from three rats each from the DIO and normal groups were selected. We extracted and purified total RNA from these samples using the miRNeasy Mini Kit (QIAGEN, GmBH, Germany) according to the manufacturer's instructions and calculated the RIN number to assess the integration of RNA using an Agilent Bioanalyzer 2100 (Agilent Technologies, Santa Clara, CA, USA).

## LncRNA microarray experiments

We performed the microarray analysis using the Rat Genome Oligo nucleotide 4,644 k Microarrays (Agilent, Santa Clara, CA, USA) at the Shanghai Biotechnology Corporation (SBC, Shanghai, China). We amplified and labeled the total RNA using the Low Input Quick Amp WT Labeling Kit (Cat.# 5190-2943, Agilent Technologies, Santa Clara, CA, USA), according to the manufacturer's instructions; the labeled cRNAs were purified using the RNeasy mini kit (Cat.# 74106, QIAGEN, GmBH, Germany). Based on the instructions in the Agilent microarray supporting kit for the Hybridization Oven (Cat.# G2545A; Agilent technologies, Santa Clara, CA, USA), the conditions for hybridization were set as 65 °C at 10 rpm for 17 h, and the volume of the cRNA sample for hybridization was 1.65 $\mu$g. The slides were then washed in staining dishes (Cat.# 121, Thermo Shandon, Waltham, MA, USA) using the Gene Expression Wash Buffer Kit (Cat.# 5188–5327, Agilent Technologies, Santa Clara, CA, USA), according to the manufacturer's instructions. The information obtained from the scanner was loaded into the image analysis program, Feature Extraction software 10.7 (Agilent Technologies, Santa Clara, CA, USA), and the data were normalized

using the Quantile algorithm from GeneSpring Software 12.6.1 (Agilent technologies, Santa Clara, CA, USA).

## Bioinformatics data analysis

After the original data was normalized using the GeneSpring Software (Agilent Technologies, Santa Clara, CA, USA), we screened high-quality probes for further data analysis. We analyzed fold-change (fold-differences in expression) and used $t$-tests (Student's $t$-test) for investigating the differentially expressed genes. After the raw data from the microarray was standardized and converted to log2 values, a scatter plot was generated in a two-dimensional coordinate system. Using the online analysis software DAVID (https://david.ncifcrf.gov/), we analyzed the Gene Ontology (GO) enrichment of the differentially expressed mRNAs and their functions, based on three aspects: biological processes (BP), cellular components (CC), and molecular functions (MF). The log10 values ($p$-value) denote enrichment scores and represent the significance of the GO term enrichment among the differentially expressed genes (DEGs). We also performed KEGG pathway analysis to reveal pathway clusters covering the DEGs; here, the log10 values ($p$-value) denote the enrichment score and represent the significance of the pathway correlations.

## Quantitative reverse transcription-polymerase chain reaction (RT-PCR) analysis

RT-PCR was used to confirm the lncRNA expression profile data obtained from the microarray. Total RNA was isolated using the Trizol reagent (Life Technologies). Single-stranded cDNA was prepared from 2 μg of total RNA, according to the manufacturer's instructions (Promega, USA), and the lncRNA expression was measured using quantitative PCR using SYBR Premix ExTaq. Two microliters of each cDNA was subjected to PCR amplification using primers specific for CUST_2117_PI428311958 (uc008nvu.1), CUST_4640_PI428311958 (AY621350), CUST_9613_PI428311958 (XR_009220.3), CUST_5105_PI428311958 (FQ225056), and CUST_6638_PI428311958 (BC058491). The lncRNA primers used in this study are shown in Table 1.

## LncRNA target prediction and lncRNA-mRNA co-expression network

Differentially expressed lncRNAs were selected for target prediction, as previously described (*Han et al., 2012*). We used two independent algorithms to identify the target genes. The first algorithm searched for those acting in cis. The University of California Santa Cruz (UCSC) gene annotations (http://genome.ucsc.edu/) were used to pair and visualize the lncRNAs in the UCSC genome browser. All genes transcribed within a 10-kbp window up- or downstream of the lncRNA were considered cis-target genes. The second algorithm was based on mRNA sequence complementarity and RNA duplex energy prediction; it evaluated the impact of lncRNA binding on complete mRNA molecules using the BLAST software for first-round screening. The RNAplex software was used to screen target genes in trans (*Tafer & Hofacker, 2008*), with the RNAplex parameter set as $e \leq -30$. Because the majority of identified LncRNAs functions were not clear, we established an lncRNA-mRNA co-expression network that comprised differentially expressed lncRNAs

**Table 1  Specific lncRNA primers for quantitative PCR analysis.**

| Primer name | Sequence (5′ — 3′) |
| --- | --- |
| CUST_2117_PI428311958-F | ATCCTGGGGTTTGTGACACT |
| CUST_2117_PI428311958-R | GGAAAGAGAAGCACCCATCA |
| CUST_4640_PI428311958-F | AGCAACGGGGACTACTGCT |
| CUST_4640_PI428311958-R | GTTCTTGAGGACCGCCACT |
| CUST_5105_PI428311958-F | GCAGGTGATTGGCTCCTAAGTC |
| CUST_5105_PI428311958-R | CAGATAACAGTGGGAAACGTCTACA |
| CUST_6638_PI428311958-F | CACCCTTCTCCGGACTTCCT |
| CUST_6638_PI428311958-R | GGACCCCAACACCTCTTTTCT |
| CUST_9613_PI428311958-F | CACACAAGCATCCCCACAG |
| CUST_9613_PI428311958-R | ATTGCGTGTGTATGTCTTTCCA |
| Rpl19(RAT15138)-F | TCCAAGGAGGAAGAGACCAA |
| Rpl19(RAT15138)-R | ACAAGGACGAAGGCTTGTTT |
| Gapdh-F | TGGCCTCCAAGGAGTAAGAAAC |
| Gapdh-R | GGCCTCTCTCTTGCTCTCAGTATC |

for cis- and trans-targeted mRNAs from the re-annotated Affymetrix Rats Genome Array data to reveal the connection between lncRNAs and mRNAs.

## Statistical analysis

Quantile normalization and subsequent processing of the raw data were performed using the GeneSpring Software GX 12.6.1 (Agilent technologies, Santa Clara, CA, USA). The results were reported as mean $\pm$ SEM from three independent tests. Student's $t$-test was performed using SPSS (13.0) to estimate the statistical significance of differences between groups. $P < 0.05$ was considered statistically significant.

## RESULTS

### Effects of high-fat diet on glycolipid metabolism in SD rats

We compared the fasting levels of blood glucose, low-density lipoprotein (LDL), high-density lipoprotein (HDL), and triglycerides (TG), as well as the body weight of DIO and normal rats. The results showed that the levels of HDL, LDL, and TG in the DIO group were not significantly higher than those in the control group. However, the fasting blood glucose level and body weight of the DIO rats were significantly higher than those of the control group (Fig. 1, Data S1).

### Effects of high-fat diet on sperm motility and testicular morphological structure in SD rats

Semen analysis show that the sperm concentration and motility in the DIO group rats ($53.63 \pm 13.82 * 10^6$/ml, $44.33 \pm 7.81\%$, respectively) was significantly lower than that in the normal control group ($92.18 \pm 6.99 * 10^6$/ml, $69.14 \pm 2.46\%$, respectively) ($p < 0.05$). In the normal control group, the testicular tissue showed that the sperm cells of the seminiferous tubules were normal and the sperm cells were arranged closely. The cell structure was clear at all stages, and a large number of mature spermatozoa were found in the lumen. At

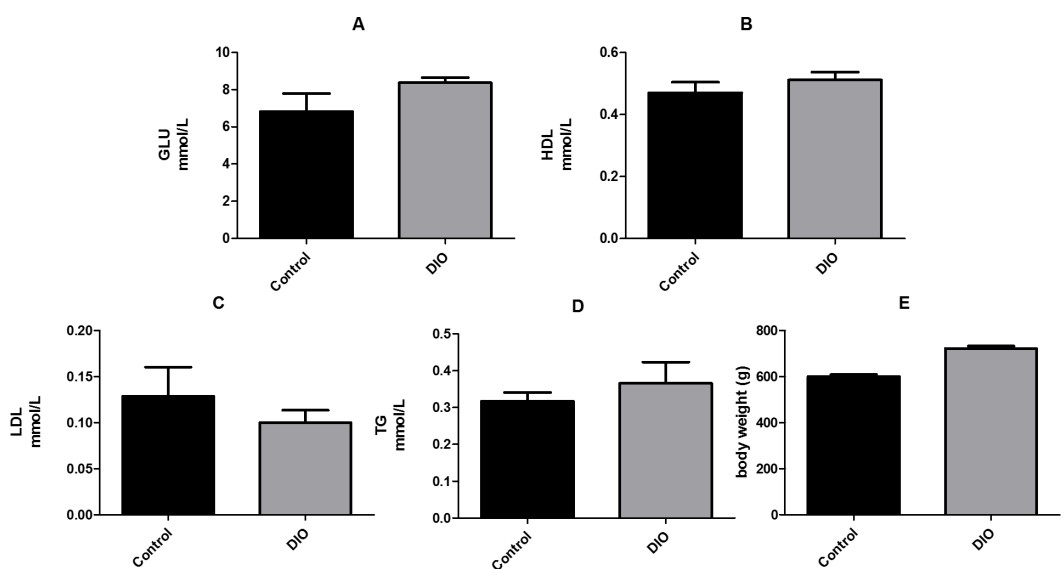

**Figure 1** **Effect of high fat diet induced obesity (DIO) on GLU, HDL, LDL and TG. Data are expressed as mean ± SEM.** ** $p < 0.01$ compared with the control group, $n = 7$.

the same time, the testicular interstitial cells between seminiferous tubules can be clearly observed round and distributed in clusters; compared with the control group, the number of sperm cells of obese rats induced by high fat diet was decreased and arranged disorderly in the testes. The number of mature spermatozoa decreased significantly, and the sperm cells were found fallen in clusters in the lumen, sperms appeared deformity, and substantial cells were significantly reduced (Fig. 2).

## Expression levels of lncRNA and mRNA in the spermatozoa of DIO and normal rats

Rat lncRNA Microarray (V6.0) is capable of detecting 23,260 lncRNAs and 26,623 mRNAs. In the present study, 9,843 lncRNAs and 23,183 mRNAs were detected in the sperm samples. After microarray scanning and normalization, 973 lncRNAs and 2,994 mRNAs were found to be differentially expressed, with fold change ≥2.0 and $P < 0.05$. Among these, 457 and 516 lncRNAs were up-regulated and down-regulated, respectively, while, 1,316 and 1,678 mRNAs were up-regulated and down-regulated (fold change ≥ 2.0 and $P < 0.05$), respectively, in the sperms of the DIO rats, compared with controls. Thirty-three lncRNAs displayed fold change >10, among which 16 were up-regulated and 17 were down-regulated (Data S2; Table 2). CUST_6253_PI428311958 (fold change: 29.3) was the most up-regulated lncRNA, while CUST_3471_PI428311958 (fold change: 36.8) was the most down-regulated lncRNA in the sperms of DIO rats, compared with controls. Twenty-five mRNAs displayed fold change >15, among which 9 were up-regulated and 16 were down-regulated (Data S3; Table 3). Visualization using scatter plots showed significant variations in the expression levels of lncRNAs and mRNAs (Fig. 3).

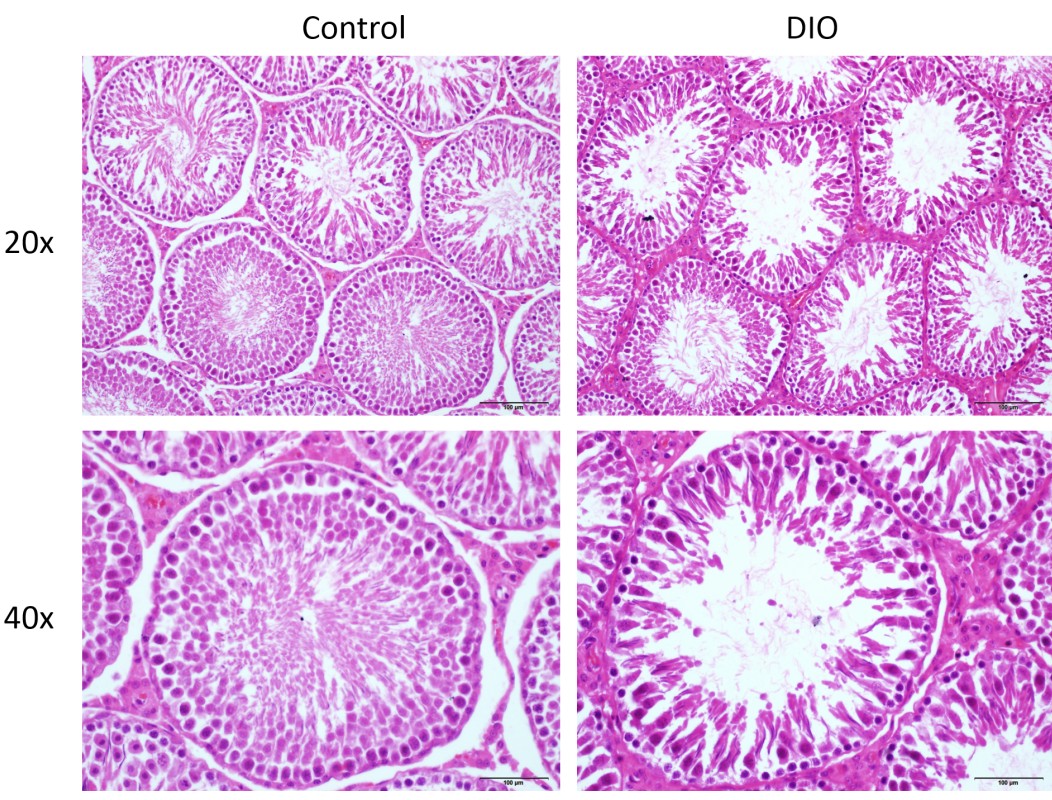

|  | Control | DIO |
|---|---|---|
| 20x | | |
| 40x | | |

**Figure 2** HE staining in the testis of control and high fat diet fed SD rat (original magnification, 20× and 40×).

## GO and pathway analysis

LncRNAs are known to be involved in the function of the corresponding mRNA gene, and the mRNAs found to be significantly differentially expressed based on the GO enrichment analysis could reveal differences in the regulation of lncRNAs. In this study, prediction terms with $p$-value <0.01 were selected and ranked based on the enrichment factor ([Count/ Pop. Hits]/[List. Total/Pop. Total]) or enrichment score ($-\log10$ [p-value]). From our data, 263 BP terms, 39 CC terms, and 40 MF terms were found ($p < 0.01$) in the differentially expressed mRNAs (Data S4). Here, we showed that the GO terms with the top 10 enrichment scores and those with the top 30 enrichment factors for differentially expressed mRNAs were associated with biological processes. Further, the cellular components were most relevant to an extracellular matrix component, the extracellular matrix. In addition, the GO terms for molecular function were correlated with growth factor binding, which is most important for sperm formation and semen quality (Figs. 4A– 4C).

KEGG pathway analysis of mRNAs that were significantly differentially expressed was performed to detect the pathways and molecular interactions associated with these genes. A total of 38 important KEGG pathways were found with $P$ value <0.05, and they were ranked based on their enrichment scores ($-\log10$ [p-value]) (Data S5). Our data showed that the pathways with the top 11 enrichment scores were associated with mRNAs. The metabolic pathway was the top pathway in protein-coding genes such as

**Table 2  Differentially expressed lncRNAs (Foldchange > 15, $P \leq 0.05$).**

| Probe name | Fold change | Regulation | Probe name | Fold change | Regulation |
|---|---|---|---|---|---|
| CUST_6253_PI428311958 | 29.30 | up | CUST_3471_PI428311958 | 36.85 | down |
| CUST_8359_PI428311958 | 23.57 | up | CUST_458_PI428311958 | 19.62 | down |
| CUST_8135_PI428311958 | 21.67 | up | CUST_5519_PI428311958 | 18.85 | down |
| CUST_787_PI428311958 | 18.58 | up | CUST_2111_PI428311958 | 17.15 | down |
| CUST_5947_PI428311958 | 16.69 | up | CUST_9135_PI428311958 | 15.42 | down |
| CUST_446_PI428311958 | 16.08 | up | CUST_8347_PI428311958 | 14.33 | down |
| CUST_5041_PI428311958 | 14.26 | up | CUST_2931_PI428311958 | 13.49 | down |
| CUST_305_PI428311958 | 13.14 | up | CUST_4808_PI428311958 | 12.80 | down |
| CUST_6233_PI428311958 | 12.77 | up | CUST_6638_PI428311958 | 12.14 | down |
| CUST_7638_PI428311958 | 12.44 | up | CUST_4899_PI428311958 | 12.02 | down |
| CUST_279_PI428311958 | 11.95 | up | CUST_396_PI428311958 | 11.75 | down |
| CUST_3553_PI428311958 | 11.51 | up | CUST_8936_PI428311958 | 11.45 | down |
| CUST_3255_PI428311958 | 10.33 | up | CUST_6637_PI428311958 | 11.32 | down |
| CUST_7370_PI428311958 | 10.24 | up | CUST_8558_PI428311958 | 11.27 | down |
| CUST_2918_PI428311958 | 10.23 | up | CUST_6993_PI428311958 | 11.18 | down |
| CUST_677_PI428311958 | 10.17 | up | CUST_4393_PI428311958 | 10.99 | down |
| | | | CUST_8371_PI428311958 | 10.60 | down |

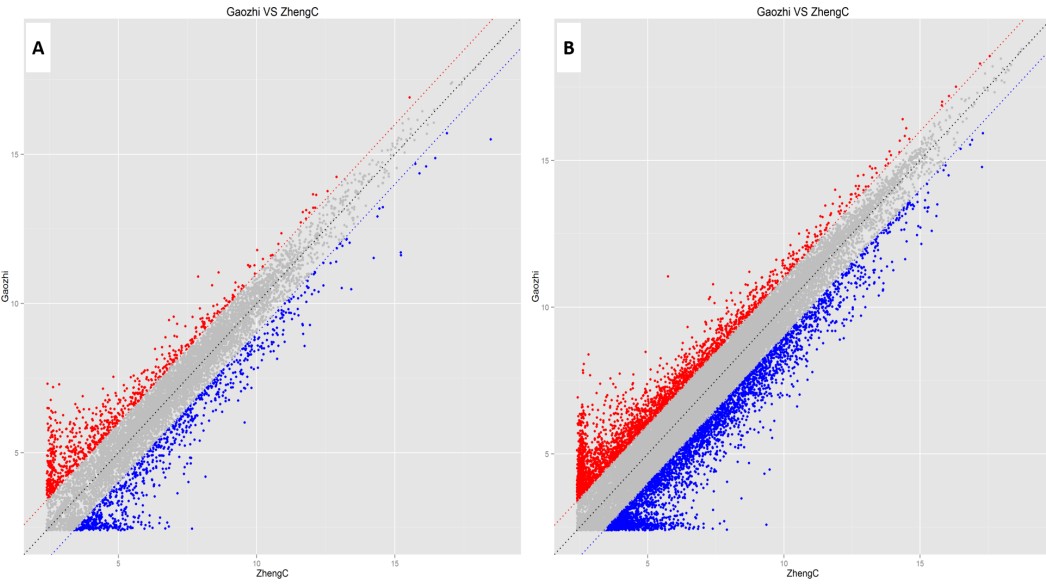

**Figure 3  Scatter plots assessing the variation in expression of lncRNAs (A) and mRNAs (B) in the two compared groups.** $X$- and $Y$-axes represent averaged normalized signal values of the microarray samples of the control and experimental groups. LncRNAs and mRNAs below the blue line and above the red line showed greater than 2.0-fold variation in expression between the two groups.

**Table 3** Differentially expressed mRNAs (Foldchange > 15, $P \leq 0.05$).

| ProbeName | foldchange | Regulation | GeneSymbol |
|---|---|---|---|
| RAT03432 | 47.38 | up | Pga5 |
| RAT02407 | 42.69 | up | Gabrg1 |
| RAT07570 | 39.46 | up | Spink3 |
| RAT02312 | 35.21 | up | Hao2 |
| RAT05103 | 27.80 | up | Ccl21 |
| RAT08360 | 22.52 | up | RGD1563982 |
| RAT25028 | 16.13 | up | B3galt6 |
| RAT15628 | 15.98 | up | Hgd |
| RAT10614 | 15.55 | up | Mrpl35 |
| RAT08939 | 108.82 | down | Olr705 |
| RAT05594 | 44.67 | down | Kcna2 |
| RAT27248 | 34.14 | down | Sertm1 |
| RAT05047 | 32.43 | down | Lrrn2 |
| RAT05303 | 32.12 | down | Sbsn |
| RAT03054 | 30.99 | down | RGD1560244 |
| RAT20303 | 30.99 | down | Olr440 |
| RAT28310 | 23.55 | down | Ptger3 |
| RAT04407 | 21.45 | down | Zp2 |
| RAT08234 | 20.19 | down | Olr96 |
| RAT21480 | 19.74 | down | Tnfrsf26 |
| RAT09493 | 18.27 | down | Itga7 |
| RAT18225 | 16.53 | down | Slc16a5 |
| RAT29116 | 16.21 | down | Adam26b |
| RAT04279 | 15.53 | down | Mpeg1 |
| RAT07795 | 15.51 | down | Fetub |

"mucin type O-glycan biosynthesis," "tyrosine metabolism," "protein digestion and absorption," "complement and coagulation cascades," "drug metabolism-cytochrome P450," "peroxisome," and "carbon metabolism." The result suggested that these pathways might contribute significantly to the pathogenesis and development of DIO-associated male infertility (Fig. 5).

## Verification of the microarray data by RT-PCR

We randomly selected five dysregulated lncRNAs, including both up-regulated (CUST_2117_PI428311958, CUST_4640_PI428311958, CUST_9613_PI428311958, CUST_5105_PI428311958) and down-regulated (CUST_6638_PI428311958) ones, for verification with sperm samples from three other rats, using GAPDH and RPL19 as the internal standards. The dissolution curve analysis showed a single peak, indicating that the specificity of PCR amplification and sample triplet repeat was satisfactory. The results from the RT-PCR and microarray were consistent with each other. Thus, the results of qRT-PCR verified the accuracy of the microarray data, providing valid evidence that lncRNAs might play an important role in the pathogenesis of male infertility caused by DIO (Fig. 6).

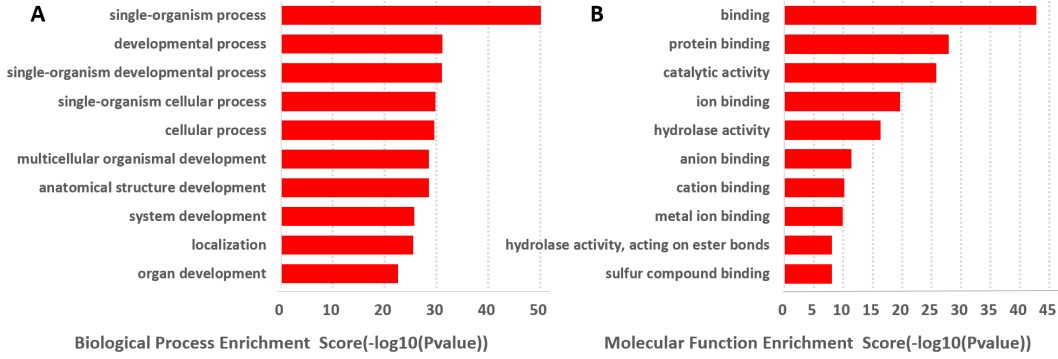

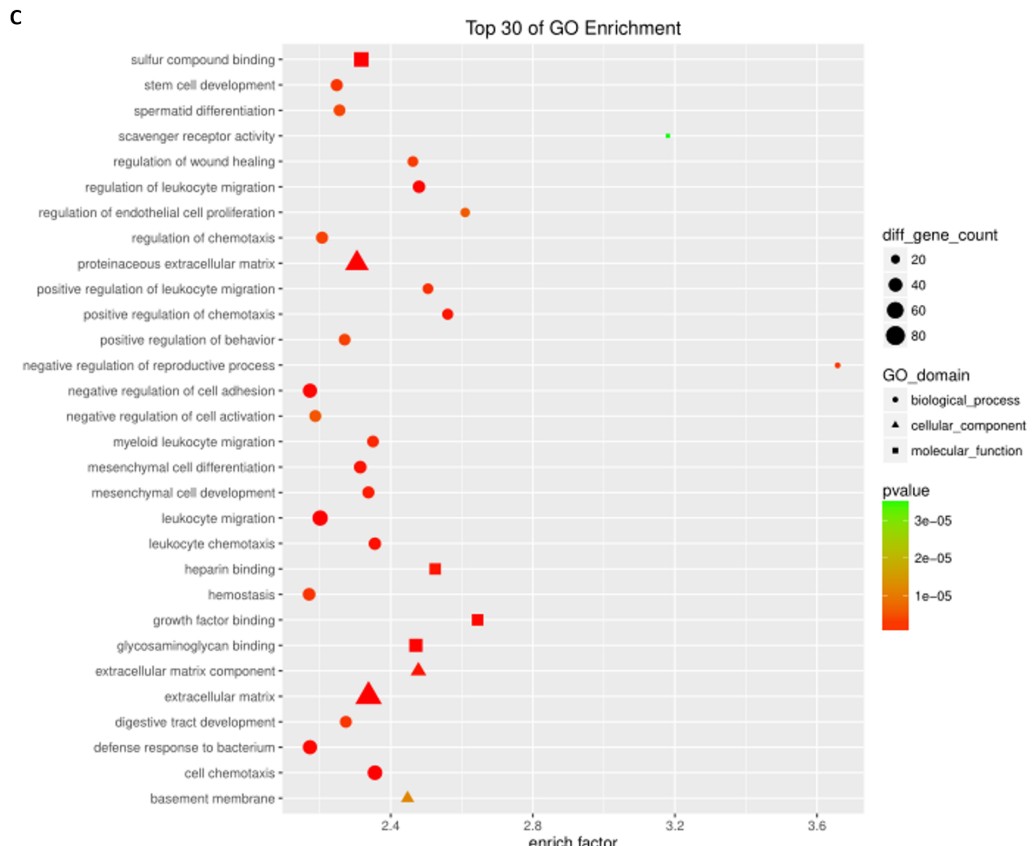

**Figure 4** **GO analysis of differentially expressed mRNAs.** Top 10 GO terms BP (A) and MF (B) ranked by enrichment scores are shown. (C) GO annotations of differentially expressed mRNAs with top 30 enrichment factors ([Count/Pop. Hits]/[List. Total/Pop. Total]) covering domains of biological processes (BP, circles), molecular functions (MF, squares), and cellular components (CC, triangles). Size represents the number of enriched genes and color indicates the degree of enrichment.

## Coding-non-coding gene network

We established a form that included the differential expression lncRNAs for cis- (Data S6) and trans- (Data S7) targeted coding genes from the re-annotated Affymetrix Rat Genome Array data. In addition, we selected four up-regulated lncRNAs, CUST_2117_PI428311958 (uc008nvu.1), CUST_4640_PI428311958 (AY621350), CUST_5105_PI428311958 (FQ225056), and CUST_6805_PI428311958 (FQ212903), and two down-regulated

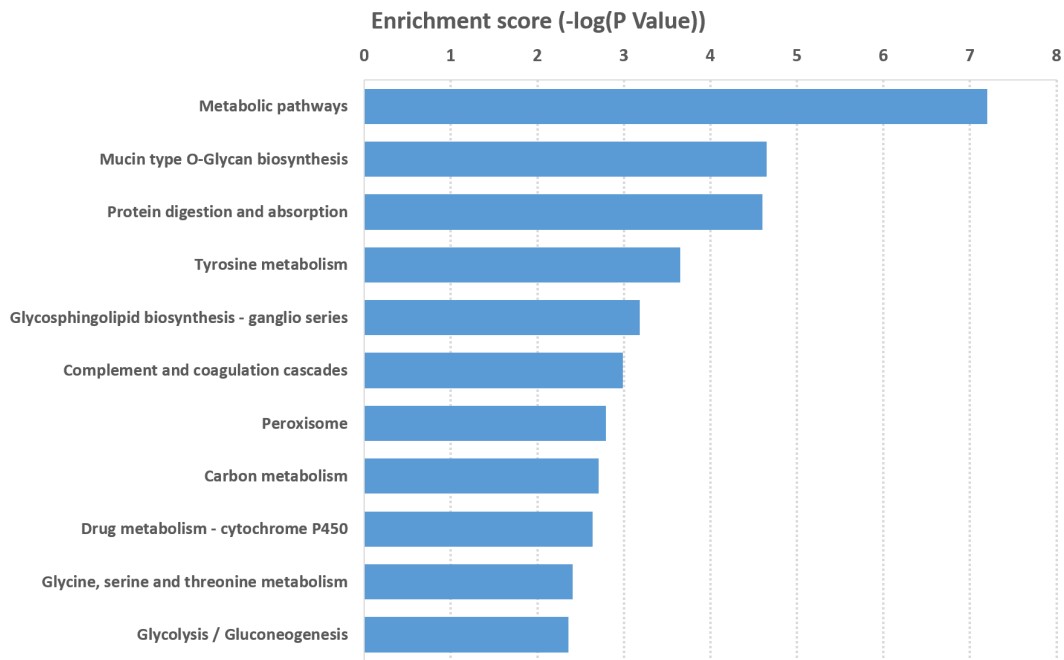

**Figure 5 KEGG pathway analysis of differentially expressed mRNAs.** Analysis of the enrichment scores (−log 10 [*P* value]) of differentially expressed mRNAs with top 11 terms.

lncRNAs, CUST_1425_PI428311958 (uc008cdl.2) and CUST_6637_PI428311958 (AF139830), for cis- and trans-targeted gene prediction. We then established an lncRNA-mRNA network. Through target gene prediction, target genes of the 26 aforementioned mRNAs were detected (Fig. 7).

## DISCUSSION

Infertility refers to the condition suffered by a couple who could not get pregnant despite one year of healthy sexual life without the use of contraceptive measures. The incidence of infertility has been significantly increasing, and male infertility accounts for 25–30% of it (*Jensen et al., 2004*). Studies have shown that male fertility may be severely affected by changes associated with obesity, type II diabetes, and metabolic syndrome (*Hammoud et al., 2006*; *Pasquali, 2006*; *Ghanbari et al., 2015*). Obesity and male infertility are known to be closely related, with the incidence of infertility in obese men being significantly higher than that in normal males (*Sermondade et al., 2013*). The effect of obesity on male reproductive capacity is complex and multifaceted. Studies have shown that obesity can cause sexual retardation (*Lee et al., 2010*), while increased body mass index (BMI) has been shown to have a negative impact on the levels of luteinizing hormone, testosterone, gonadotropin, sex hormone binding protein, and estradiol in men (*Hart et al., 2015*; *Fui, Dupuis & Grossmann, 2014*). Some studies have also shown that obesity can cause erectile dysfunction, affecting the volume, concentration, activity, and count of sperms. Obesity is also closely associated with increased sperm DNA damage (*Pan et al., 2015*; *Magnusdottir et al., 2005*; *Dupont et al., 2013*). Thus, there is a considerable amount of evidence for a strong

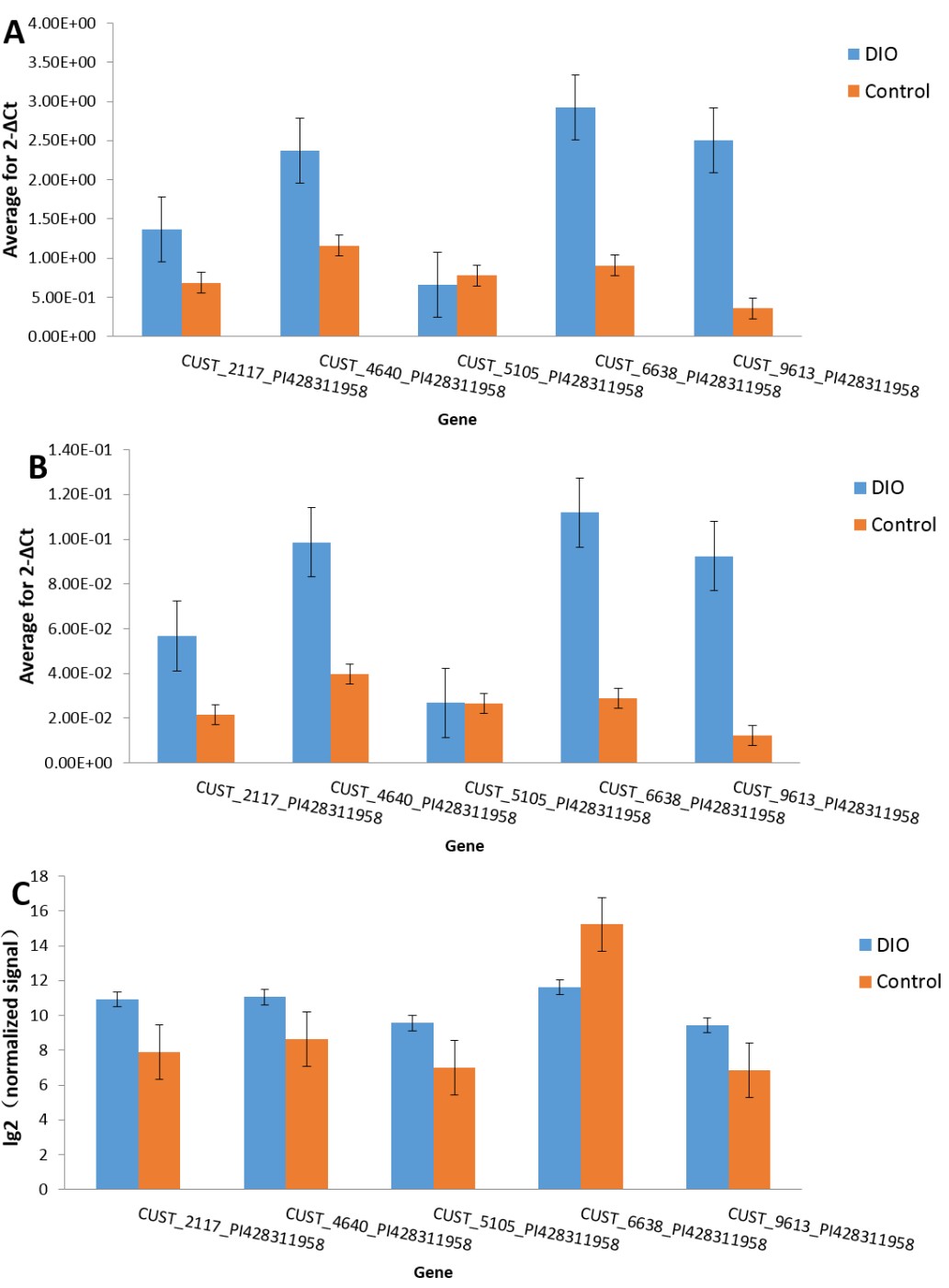

**Figure 6 Validation of microarray data by qRT-PCR.** Comparison of the results of qRT-PCR and microarray for lncRNAs. Results obtained with these two methods were consistent with each other. (A) Gapdh; (B) RPL19; (C) Array.

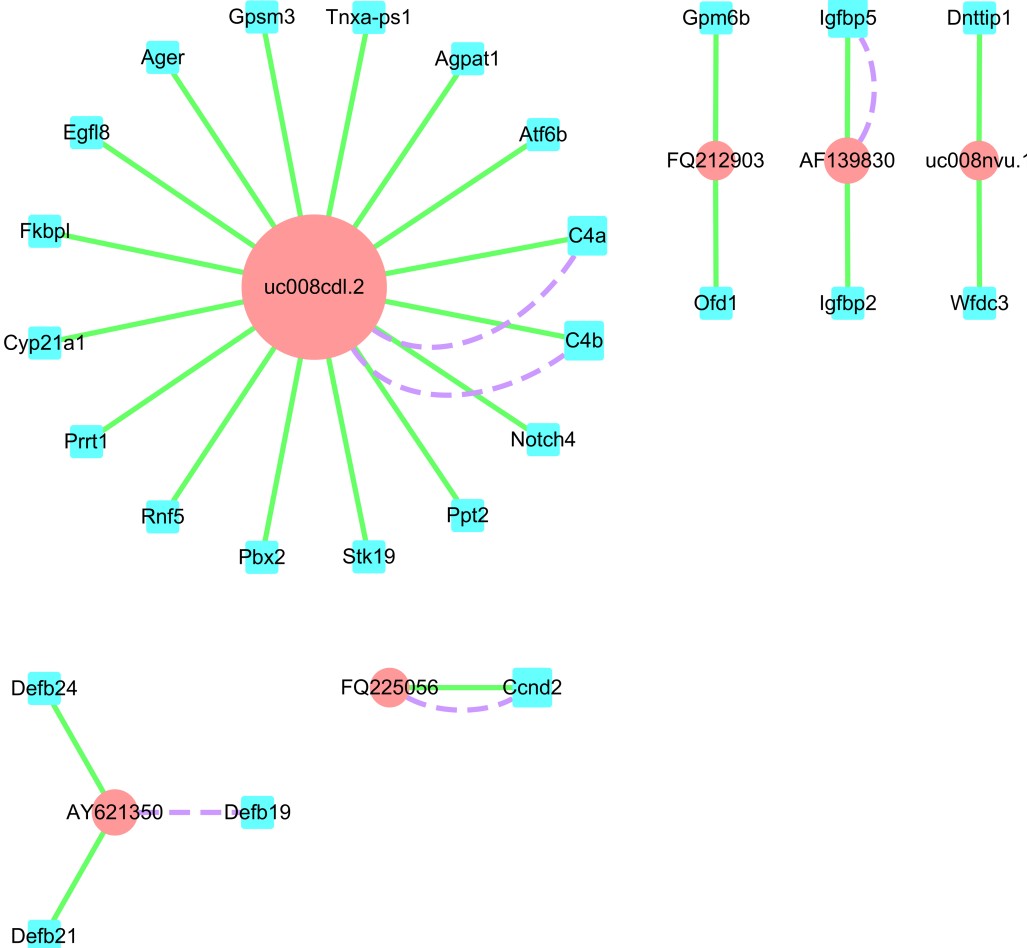

**Figure 7 LncRNA-mRNA network.** Blue square nodes and pink round nodes represent mRNAs and lncRNAs, respectively; purple dashed lines and blue solid lines between two nodes represent trans- and cis-targets, respectively. The size of the points indicates the number of targets associated with the lncR-NAs.

correlation between obesity and male infertility. Therefore, studying the effect of obesity on the mechanism and pathophysiological process of male infertility has high clinical value.

LncRNAs generally have no coding potential and are longer than 200 nt. Originally, lncRNAs were considered the "noise" of genome transcription, with no biological function (*Gordiiuk, 2014*). However, many studies have recently demonstrated that lncRNAs play important roles in regulating gene expression by epigenetic, transcriptional, and post-transcriptional regulation; they have also been shown to affect cell proliferation, differentiation, metabolism, and apoptosis (*Caley et al., 2010*; *Maass, Luft & Bähring, 2014*). LncRNAs were found to be differentially expressed in sperm samples from obese and normal subjects, indicating that it might be a target for therapy against obesity-associated male infertility. In this study, we confirmed that lncRNAs were differentially expressed in sperm from DIO and normal rats. We also explored the effect of obesity on reproduction at the molecular level and the effect of lncRNAs on obesity and male reproduction.

In this study, we performed a comprehensive analysis of dysregulated lncRNAs by comparing the transcriptome profiles of sperm samples from obese and normal rats. A total of 973 lncRNAs were discovered. Among these, 457 were up-regulated and 516 were down-regulated; we extracted their general features. We selected three up-regulated (CUST_2117_PI428311958, CUST_9613_PI428311958, and CUST_5105_PI428311958) and two down-regulated (CUST_6638_PI428311958, CUST_396_PI428311958) lncRNAs for verification using qRT-PCR. The results of the qRT-PCR analysis were consistent with the microarray results, indicating that the microarray data was reliable. Thus, our study provided a comprehensive understanding of the role of lncRNAs in DIO-induced male infertility; our results could help understand the epigenetic effects of lncRNAs on male infertility in obese patients.

GO term enrichment analysis was performed to identify biological processes, cellular components, and molecular functions associated with the differentially expressed lncRNAs. We found that the differentially expressed lncRNAs were highly enriched in functions related to biological process such as negative regulation of reproductive processes and regulation of endothelial cell proliferation; cell components such as extracellular matrix component, basement membrane, and scavenger receptor activity; and molecular functions such as growth factor binding. All these were closely associated with male infertility. In addition, pathway analysis showed a significant change in metabolic pathways, mucin type-O-Glycan biosynthesis, protein digestion and absorption, tyrosine metabolism, glycosphingolipid biosynthesis-ganglio series, and cytokine-cytokine receptor interaction. These results suggested that metabolic, endocrine, and other abnormalities might affect obese patients.

In this study, we found many dysregulated lncRNAs in the sperm samples of rats with DIO, and predicted their corresponding mRNAs through cis- and trans-targeting. For example, among the detected mRNAs in cis, up-regulated lncRNA CUST_2117_PI428311958 (uc008nvu.1; fold change: 8.12) was predicted to act on Wfdc3 (Data S6), which is a WFDC type serine protease inhibitor located on human chromosome 20. Studies have shown that Wfdc3 is highly expressed in the epididymis, sperm, testes, and other male reproductive organs. It therefore plays a potential role in male fertility (*Jalkanen, Kotimaki & Poutanen, 2006*). Wfdc3 was the predicted target gene for hsa-miR-487a, which has been detected in the microRNA expression profile of the sperm of patients with asthenospermia (*Landgraf et al., 2007*). However, the role of Wfdc3 in the reproductive process has not been extensively studied, and the association of mutations in the WFDC protease inhibitor gene with male infertility need to further studied. Furthermore, the microarray analysis had predicted that CUST_5105_PI428311958 (FQ225056; fold change: 5.87) would act both in cis (Ccnd2) and trans (Ccnd2) (Datas S6 and S7). Ccnd2 is associated with cellular regulation, and its dysregulated expression could lead to abnormal cell proliferation (*Dong et al., 2010*). A previous study had confirmed that the risk of type 2 diabetes was halved by the presence of a low-frequency allele in lncRNA-CCND2 that promoted insulin secretion (*Yaghootkar et al., 2015*). Thus, CUST_2117_PI428311958 and CUST_5105_PI428311958 could be mediators for the occurrence and progression of obesity-associated male infertility. However, due to the lack of known function of lncRNAs,

lncRNA-mRNA interactions should be studied in detail. It would be particularly important to improve our understanding of the mechanisms behind lncRNA-associated diseases and available techniques to diagnose and prevent them.

In this study, we constructed and analyzed the expression patterns of mRNAs and lncRNAs in DIO and normal rats. Our results revealed many important lncRNAs, whose expression levels affected the development of obesity. Further studies will be necessary to investigate the molecular mechanisms of action of specific lncRNAs, which could help in the exploration of novel therapeutic targets in DIO-associated male infertility.

## CONCLUSIONS

In summary, we detected the abnormal expression of lncRNAs and mRNAs in the sperm samples of DIO rats, and analyzed the potential roles of mRNAs through bioinformatics. The GO term enrichment analysis showed that the function most highly enriched was related to negative regulation of reproductive processes. Pathway analysis showed that metabolic pathways might be related to the obesity-induced decline in male fertility. Our results would be helpful for future studies that investigate the molecular role of lncRNAs in DIO-associated male infertility. We provided experimental data on male infertility caused by obesity, and the lncRNA expression profile that we constructed could contribute to future studies that investigate the molecular functions of lncRNAs in obesity-associated decrease in male fertility.

## ACKNOWLEDGEMENTS

The authors would like to acknowledge Bo Hao, Shanghai, China, were the microarray experiments were performed.

### Funding

This study was supported by grants from the National Natural Science Foundation of China (NSFC30770247) the key drug development Programme of MOST (20122X09103201-005), the Production and Research Joint Cultivation Project (1000062520181) and the International Cooperation Projects of MOE (2011DFA30920). The funders had no role in study design, data collection and analysis, decision to publish, or preparation of the manuscript.

### Grant Disclosures

The following grant information was disclosed by the authors:
National Natural Science Foundation of China: NSFC30770247.
Key drug development Programme of MOST: 20122X09103201-005.
Production and Research Joint Cultivation Project: 1000062520181.
International Cooperation Projects of MOE: 2011DFA30920.

## Competing Interests

The authors declare there are no competing interests.

## Author Contributions

- Tian An conceived and designed the experiments, performed the experiments, analyzed the data, wrote the paper, prepared figures and/or tables, reviewed drafts of the paper.
- Hui Fan conceived and designed the experiments.
- Yu F. Liu conceived and designed the experiments, analyzed the data.
- Yan Y. Pan performed the experiments.
- Ying K. Liu contributed reagents/materials/analysis tools.
- Fang F. Mo and Dan D. Zhao prepared figures and/or tables.
- Yu J. Gu performed the experiments, analyzed the data.
- Ya L. Sun and Qiu L. Wang analyzed the data.
- Na Yu and Yue Ma contributed reagents/materials/analysis tools.
- Chen Y. Liu and Zheng Y. Li performed the experiments.
- Fei Teng analyzed the data, help to feed animals.
- Si Hua Gao conceived and designed the experiments, reviewed drafts of the paper.
- Guang J. Jiang conceived and designed the experiments, wrote the paper, reviewed drafts of the paper.

## Animal Ethics

The following information was supplied relating to ethical approvals (i.e., approving body and any reference numbers):

This research was allowed by the Animal Care Committee of the Institute of Zoology, Chinese Academy of Sciences and all rats operates were at the request of the guidelines of the Animal Care Committee.

## Field Study Permissions

The following information was supplied relating to field study approvals (i.e., approving body and any reference numbers):

Field experiments were approved by the Research Council of Beijing University of Chinese Medicine (certification number SCXK (Jing) 2011-0024).

## Data Availability

The raw data has been supplied as a Supplementary File.

## Supplemental Information

Supplemental information for this article can be found online at http://dx.doi.org/10.7717/peerj.3518#supplemental-information.

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
