# Peer review of "The difference in expression of long noncoding RNAs in rat semen induced by high-fat diet was associated with metabolic pathways"

_PeerJ, doi:10.7717/peerj.3518_

## Round 0.1 · original submission · Major Revisions

As the two reviewers commented, the manuscript is immature and contains many typos. Please revise the manuscript following all suggestions given by the reviewers.

Reviewer 1 ·

Basic reporting

In this manuscript, An et al aimed to correlate expression changes in lncRNA with those in mRNA in spermatozoa of obese rats induced by HFD in order to understand a mechanism by which obesity is a risk factor for male infertility. The authors isolated RNA from sperm supernatant fluid of obese and control rats, followed by analyses of microarrays and bioinformatics. Through the lncRNA-mRNA co-expression network analysis, the authors aimed to understand the roles of some lncRNA whose expression levels were changed in the obese condition. While this attempt is of interest, there are multiple technical and biological issues that make the data difficult to interpret properly.

Experimental design

1. While the authors stated that HFD-induced obese rates were used for the analysis, the rats did not show typical symptoms of obesity such as hyperlipidemia except for slight induction of fasting blood glucose levels. Did these rats actually become obese compared to their controls? Body weight information of each mouse should be included in the manuscript (e.g., Fig.1 and S1) to advocate your animal model as a HFD-induced obesity. What is a formula of normal diet used as a control?
2. In this study, spermatozoa were placed in the cultures of preheated human tubal and centrifuged to collect the supernatant fluid. This procedure needs to be described in more detail. In this procedure, functional sperms can be collected, however, damaged and motionless sperm may not. Obesity is associated with hypogonadism, impaired spermatogenesis, and sperm functions. Therefore, the procedure may result in technical biases of sperm collection in this study. What are fundamentals of the collected sperm, e.g, number, activity, and shape, used for RNA analysis? If there is a bias of sperm collection, it certainly affects lncRNA and mRNA expression patterns and the lncRNA-mRNA network analysis.

Validity of the findings

1. The authors selected 5 out of 973 lncRNAs for q-PCR assay to verify the microarray data. Even if there are in good consistency between q-PCR and microarray data with the sample size, it is scientifically difficult and inappropriate to state that the q-PCR results verify the veracity of microarray data results due to the smaller sample power. In addition, there is no statistic analysis in Figure 5 and CUST_5105 and 6638 show different trends of expression pattern between q-PCR and microarray data.
2. Regarding Figure 6, it is difficult to interpret and understand the data. There are only limited information about a purpose of this analysis and how this was done. While there are some descriptions about your interpretation in “Discussion,” it still leaves ambiguities in interpreting the results. For instance, the authors state that “CUST_5105_PI428311958 (fold change: 5.87) microarray was predicted to act both in cis (Ccnd2) and in trans (Ccnd2) (Supplementary Data Table S6, S7)” in line 250. However, it is difficult for reviews/readers to find such a correlation in the vast supplementary data sets.

Additional comments

Overall, due to uncertainty of the experimental conditions, I find this very confusing and I am afraid that the readers will find it confusing as well.

Reviewer 2 ·

Basic reporting

The English language should be improved, and the manuscript, in which a lot of typos and grammar mistakes are found, must be sent to the editing services before being published. In addition, some abbreviations, such as BP, CC, MF, SD, and DEGs, are not explained.

Regarding to the LncRNA-mRNA co-expression network analysis, it would be necessary to describe in details what lncRNA-mRNA network means. What are cic- and trans-targeted mRNAs? To understand the network analysis well, a description about the mechanism how lncRNAs control gene expression via epigenetic, transcriptional and post-transcriptional regulation would be included in the Introduction and Result sections.

Experimental design

In the Bioinformatical analysis of microarray data, mRNA that have low intensities are defined as absences, but mRNAs having low intensities seem to be still shown in Figure 2. How many mRNAs are present? It is necessary to describe more detailed method of the analysis.

In line 80, to validate the effect of high-fat diet feeding, could you show body weight data?

In line 141, would you describe how many probes are put for lncRNAs and mRNA in the chip and how many lncRNAs and mRNAs are present in this tissue?

In Figure 3C, it is difficult to see GO annotations. Could you enlarge the font size?

In Figure 5C, error bars from the triplicates can be added to bar graphs.

Could you explain why the 5 lncRNAs are selected for the qPCR and network analyses.

In Tables 2 and 3, several probes are identified as NA. What does NA mean? These NA genes should be excluded for the subsequent analyses.

It would be interesting to compare the data between obese rats and mice from data published by the authors.

It would be necessary to see whether feeding of high-fat diet affects spermatogenesis and fertility, such as sperm volume, sperm concentration, activity of sperm, histology of testis, etc.

Validity of the findings

Authors previously provided the expression profiles from diabetec mice sperm. It is necessary to add the rationale why obese rats are used.

The authors concluded that “GO term enrichment analysis showed that the most highly enriched in function is related to negative regulation of reproduction process”. However, the Figure 3 identifies single-organism process, developmental process, sulfur compound binding and stem cell development. It would be necessary to change the conclusion.

Additional comments

The manuscript by An and colleagues shows the expression profiles of lncRNAs and mRNAs from died-induced obese rats, using microarray. Although this group have previously provided the information from obese mice, the expression profiles of obese rats have not been previously investigated. Additional analyses and experiments would help to strength this study. The English language should be improved and the manuscript must be sent for the editing services to correct many typos and grammar mistakes.

---

## Round 0.2 · accepted · Accept

Your manuscript is now accepted for publication. Although it is not mandatory, I appreciate it if you could include your response to the reviewer's minor comments in your final manuscript.

Best,
Tadasih

Reviewer 2 ·

Basic reporting

The authors addressed most of the reviewers’ concerns by adding additional data and modifying the text. The manuscript has been significantly improved. I have only fairly minor comments to improve the manuscript.

1. Background about the lncRNA-mRNA network should be added into the introduction to understand significance of the network analysis.

Experimental design

1. The author answered our concern about the way to select five lncRNAs. However, there is a discrepancy between the text and the point-by-point responses. Did the author selected 5 lncRNAs randomly, according to the fold change values and the molecular functions, or due to the limitations of research funding?

Validity of the findings

No comment.

Additional comments

No comment.